# New Onset Atrial Fibrillation in STEMI Patients: Main Prognostic Factors and Clinical Outcome

**DOI:** 10.3390/diagnostics13040613

**Published:** 2023-02-07

**Authors:** Beatrice Dal Zotto, Lucia Barbieri, Gabriele Tumminello, Massimo Saviano, Domitilla Gentile, Stefano Lucreziotti, Loredana Frattini, Diego Tarricone, Stefano Carugo

**Affiliations:** 1Department of Cardio-Thoracic-Vascular Diseases, Foundation IRCCS Ca’ Granda Ospedale Maggiore Policlinico, 20122 Milan, Italy; 2UOC Cardiology, ASST Santi Paolo e Carlo, 20142 Milan, Italy; 3Department of Clinical Sciences and Community Health, University of Milan, 20122 Milan, Italy

**Keywords:** atrial fibrillation, STEMI patients, anticoagulant, stroke, bleeding

## Abstract

The indications for the treatment of patients with known atrial fibrillation (AF) undergoing percutaneous coronary intervention (PCI) are clear, while less is available about the management of new-onset AF (NOAF) during ST-segment elevation myocardial infarction (STEMI). The aim of this study is to evaluate mortality and clinical outcome of this high-risk subgroup of patients. We analyzed 1455 consecutive patients undergoing PCI for STEMI. NOAF was detected in 102 subjects, 62.7% males, with a mean age of 74.8 ± 10.6 years. The mean ejection fraction (EF) was 43.5 ± 12.1% and the mean atrial volume was increased (58 ± 20.9 mL). NOAF occurred mainly in the peri-acute phase and had a very variable duration (8.1 ± 12.5 min). During hospitalization, all the patients were treated with enoxaparin, but only 21.6% of them were discharged with long term oral anticoagulation. The majority of patients had a CHA2DS2-VASc score >2 and a HAS-BLED score of 2 or 3. The in-hospital mortality was 14.2%, while the 1-year mortality was 17.2% and long-term mortality 32.1% (median follow-up 1820 days). We identified age as an independent predictor of mortality both at short- and long-term follow-ups, while EF was the only independent predictor for in-hospital mortality and arrhythmia duration for 1-year mortality. At the 1-year follow-up, we recorded three ischemic strokes and no bleeding complications.

## 1. Introduction

Atrial fibrillation (AF) nowadays is the most frequent arrhythmia, especially among the elderly population and coronary artery disease (CAD) still represents the first cause of death in Western countries [1,2]. Mortality, primarily in ST-segment elevation myocardial infarction (STEMI), remains significant, despite major improvements both in pharmacological treatment and revascularization techniques [3]. AF has been documented with increasing frequency in patients with acute coronary syndrome (ACS), however, two distinct categories must be identified: patients with previously known AF and patients who develop new-onset AF (NOAF) concomitantly with the acute ischemic event [4]. The incidence of NOAF during ACS in the current literature is between 2.3% and 37%, whereas during it STEMI ranges from 5% to 14% [5,6]. Although many data and studies about patients with ACS and known AF are available [7,8], little is still known about the management of patients with NOAF in the acute phase of ACS and especially during the acute phase of STEMI. The etiology of NOAF seems to be multifactorial and various events can act as a trigger or participate in the onset of the arrhythmic episode: atrial ischemia and dilation, alterations of the autonomic nervous system with increased sympathetic tone and decreased vagal activity, both localized and systemic inflammation, and hormonal activation [9]. These factors lead to structural and electrical atrial remodeling that results in NOAF. Different risk factors for the development of NOAF have been identified: the most common cardiovascular risk factors (age, sex, obesity, and smoke) along with hypertension, diabetes, renal failure, COPD, previous arrhythmias, and alterations of clinical parameters such as an increase in heart rate, left atrial size and volume, and reduced ejection fraction (EF) [10,11]. These elements and other routine biomarkers were used in proven risk scores such as the ALBO (Age, Leukocytes, BNP, and Obesity) score, for the risk stratification of STEMI patients who are at a high risk of developing NOAF and GRACE (Global Registry of Acute Coronary Events) and SYNTAX (Synergy between PCI with TAXUS drug-eluting stent and Cardiac Surgery) scores [12,13,14]. When concomitant with ACS, NOAF is associated with a higher incidence of early and late adverse events: structural and functional cardiac deterioration, cardiac rhythm alterations, thrombo-embolism, ischemia, and re-infarction [15,16,17,18]. The impact of AF on in-hospital, short- and long-term follow-up mortality in STEMI patients is a controversial topic. Most studies affirm that mortality in this subgroup of patients appears to be increased, while others argue that NOAF during STEMI does not affect mortality [19]. The actual guidelines provide clear indications for both short- and long-term treatment with a combination of antiplatelet agents and oral anticoagulants (OAC) based on the relationship between ischemic and hemorrhagic risk of patients with ACS and AF. The CHA2DS2-VASc (congestive heart failure, hypertension, age >75 years, diabetes mellitus, prior stroke 2 or transient ischemic attack, or thromboembolism) and HAS-BLED Scores (hypertension, abnormal renal/liver function, stroke, bleeding history or predisposition, labile INR, elderly, or drugs/alcohol concomitantly) are strongly confirmed [20,21]. The information about the treatment of patients with transient NOAF during ACS are conversely more ambiguous and a noticeable lack of indications is particularly evident in the very high-risk subgroup of STEMI patients [10,11]. The aim of this study is therefore to evaluate mortality and clinical outcome of STEMI patients who developed NOAF during the index hospitalization.

## 2. Materials and Methods

We performed a single-center, retrospective, cohort study of 1455 patients undergoing coronary angiography and/or PCI for STEMI between October 2008 and October 2019 at San Paolo Hospital, Milan, Italy. All demographic and clinical data were prospectively collected in a specific database. The study was conducted according to GCP, institutional guidelines, national legal requirements, European standards, and the revised Declaration of Helsinki. Hypertension was defined as systolic blood pressure (BP) > 140 mmHg and/or a diastolic BP > 90 mmHg or on-treatment with antihypertensive medications. The diagnosis of diabetes was based on a previous history of diabetes treated with or without drugs, fasting glycemia >126 mg/dl or glycosylated hemoglobin >6.5%. Hypercholesterolemia was defined as a previous history of hypercholesterolemia, chronic treatment with any hypocholesterolemic agent at admission, or fasting total cholesterol >200 mg/dl. Creatinine clearance was assessed using the CKD-EPI formula [22]. The left ventricular ejection fraction (LVEF) and left atrium volume (mL) were acquired using a standard trans-thoracic echocardiography at admission.

### 2.1. Clinical Endpoint

The primary endpoints of the study were the in-hospital, 1-year, and long-term follow-up mortality. The secondary endpoints were the cerebral ischemic and hemorrhagic events. A follow-up was performed through clinical visits and telephone interviews. Death was confirmed by reviewing all the medical records including the ‘Sistema Informativo Socio-Sanitario—Regione Lombardia’, an electronic database that encompasses medical information from all the hospitals of the region.

### 2.2. NOAF Diagnosis

The presence and duration of AF were detected through a detailed review of the clinical data, electrocardiograms, and cardiac monitoring. The patients with histories of both paroxysmal and persistent AF before admission were excluded from the study.

### 2.3. Coronary Angiography

Coronary angiography was routinely performed using the Judkins technique. Significant CAD was defined as at least one >1.5 mm coronary artery showing a >70% stenosis, while severe multivessel disease was defined as at least two >2 mm vessels with >70% stenosis.

### 2.4. Stroke and Bleeding

The CHA2DS2-VASc and HAS-BLED scores were calculated for all the patients. Ischemic stroke was defined as a neurological deficit of sudden onset and lasting more than 24 h, confirmed by instrumental methods such as brain CT and MRI. Bleeding events were defined according to the criteria of the Bleeding Academic Research Consortium (BARC) [23].

### 2.5. Statistical Analysis

The statistical analysis was performed using the R 4.0 statistical package. The continuous data were expressed as mean + standard deviation (M ± SD) and categorical data as a percentage. The statistical significance of *p*-values was set at 0.05 and the confidence intervals (CI) at 95%. The multiple logistic regression analyses were performed to identify independent clinical and procedural predictors associated with in-hospital, 1-year, and follow-up mortality. The ROC curve and the area under curve (AUC) were used to validate the classification model. The survival curve was obtained by applying the Kaplan–Meier method.

## 3. Results

We analyzed a total of 1455 patients undergoing coronary angiography and/or angioplasty for STEMI. NOAF was found in 102 subjects (7%). The population (62.3% of males), with a mean age of 74.8 ± 10.6 years, had showed one or more cardiovascular risk factors: hypertension (61.8%), diabetes (21.6%), dyslipidemia (34.3%), family history of CAD (6.9%), history of CAD (2.9%), obesity (9.8%), and smoking (28.4%). Anterior STEMI was the most frequent type of AMI at admission (46%) with a Killip class 3 in 17.6% of the cases, 12.7% showed cardiogenic shock, and 5.9% presented for cardiac arrest. Renal failure and anemia were detected in 40.2% and 33.3% of the patients, respectively. The LVEF at admission was 43.5 ± 12.1% and the left atrium volume was 58 ± 20.9 mL. The coronary angiography revealed multivessel disease in 51% of the patients and single-vessel disease in 45.1%, while the coronary arteries were free from significant stenosis in 3.9%. The main cause of the ischemic events was the left anterior descendent (42.2%) followed by the right coronary artery (40.2%). NOAF mostly occurred in the peri-acute phase and was characterized by variable duration (mean 8.1 ± 12.5 min). Furthermore, 5.9% of patients developed persistent AF at discharge, in 69.6% sinus rhythm conversion was obtained with pharmacological cardioversion, in 2% using DC-shock, and spontaneous resolution occurred in 22.5%. During hospitalization, all the patients were treated with Enoxaparin (anticoagulant dose). A total of 22 patients (21.6%) were discharged from the hospital with long-term anticoagulation therapy (OAC), of which 17 (16.6%) were discharged with Warfarin and 5 (4.9%) with DOAC. A CHA2DS2-VASc score >2 was detected in 83.3% of patients, while HAS-BLED scores of 2 and 3 were the most represented (64.7%) in the population. The in-hospital mortality was 14.7% (15/102), the linear logistic regression analysis identified reduced the LVEF (OR [95% CI] = 0.993 [0.987–0.999], *p* = 0.033) and age (OR [95% CI] = 1.006 [1.000–1.013], *p* = 0.05) as the only independent risk factors. The average age in this subgroup is 80.46 ± 9.3 and mean LVEF is 35 ± 13.9%. The patients who died at the 1-year follow-up were 15/87, with a mortality of 17.2%. At the 1-year follow-up, three cerebral ischemic events (3.4%) were detected only among patients without OAC and two of them died. Conversely, no hemorrhagic events were detected. In this case, age (OR [95% CI] = 1.019 [1.011–1.026], *p* < 0.0001) and arrhythmia duration (OR [95% CI] = 1.0007 [0.999–1.001], *p* = 0.05) were detected as predictors of mortality. In the long-term follow-up (median follow-up time 1820 days, ranging from 341 to 3985 days), 28 patients died after discharge, therefore the overall mortality at follow-up was 32.1%. In this case, independent predictors were again age (OR [95% CI] = 1.021 [1.012–1.030], *p* < 0.0001) and cardiac arrest at admission (OR [95% CI] = 1.82 [1.04–3.18], *p* = 0.038). Table 1 shows clinical, procedural, arrhythmic events characteristics, and outcomes of the population. Figure 1 reports cumulative survival curve of the population. 

## 4. Discussion

Our study shows how the concomitance of NOAF during STEMI represents a very high-risk clinical condition that requires correct management both at the short and long term. Nowadays, AF is the most frequent and significant arrhythmia especially among the elderly, while ischemic heart disease is the leading cause of death in Western countries. Despite the significant progress, both in therapeutic and interventional techniques, STEMI still prompts high mortality [1,2,3,24]. The actual indications clearly delineate the management of patients with ACS and the history of AF while limited and controversial evidence is available for the patients with NOAF in the acute phase of STEMI [25,26]. The increased risk of bleeding in patients with AF undergoing PCI due to an ACS in comparison to those *with chronic coronary syndrome, is quite clear* [27]. In our population, the incidence of NOAF was 7% in line with the available literature [12,13]. As reported in some previous studies [28,29], we found a high mortality rate both at the short- and long-term follow-ups detecting different clinical parameters as independent predictors. Age was as an independent predictor of mortality for in-hospital, 1-year, and long-term follow-up. This result can be explained by the fragility of elderly patients, often more compromised with multiple comorbidities [9,10,11,12,13,30,31,32]. LVEF appears to be an independent predictor only for in-hospital mortality as an index of myocardial damage secondary to the acute ischemic episode [33,34].

Additionally, the RISK-PCI study [5] highlights how in this category of patients reduced LVEF has a negative prognostic value, while, in the study by Rencuzogullari et al. [11], the decreased LVEF was considered a risk factor for the development of AF. Interestingly, at the 1-year follow-up, in addition to age, we identified arrhythmia duration as the only predictor for mortality. Moreover, a total of three ischemic strokes, two of them fatal, were reported. Conversely, no hemorrhagic events were detected. This finding is justified by the linear relation between the duration of the arrhythmia and the probability of its relapse, which exposes patients to an higher risk of thromboembolic events [30]. Together with our results, several studies showed an increased risk of stroke in the STEMI patients [6,13,28,30], which is presumably related to a higher CHA2DS2-VASc Score [10,29]. Moreover, a sub-analysis of the HORIZON-AMI Trial [35] and the Portuguese National Registry [15] found not only an increase in ischemic risk but also in the hemorrhagic risk in STEMI patients, confirming in both cases a high incidence of adverse events and mortality. The HORIZON-AMI Trial analysis attributed the high rate of bleeding in their population (20.9%) to triple therapy use. This background and the high clinician awareness of the hemorrhagic risk may explain the reduced prescription of anticoagulation in the “real world” and in our population. The GARFIELD, a large “real world” register, highlights how there is a large underuse of anticoagulant therapy due to an overestimation of the hemorrhagic risk compared to thrombotic risk. Among the patients with concomitant indications to dual antiplatelet therapy, only 38% of the patient s with indication to OAC are correctly treated [36]. The relationship between thrombotic and hemorrhagic risk in our population is unbalanced due to a high CHA2DS2-VASc score compared to a generally low HAS-BLED score. It is impressive to see that even if the vast majority of patients (83.3%) have an indication to OAC (CHA2DS2-VASc Score ≥ 2), only 21.6% were treated after discharge with long-term OAC. Nevertheless, the onset of ischemic events is frequently temporally related to the suspension of this therapy following bleeding events, underlying how difficult the management of ischemic and hemorrhagic risk is [35]. A limit to these considerations is that most of the studies currently available were based on vitamin K antagonist. As reported above, ischemic events in our population only occurred within the group of patients not prescribed with OAC, causing 2 of 3 deaths detected at 1 year. The concomitant complete absence of hemorrhagic events at the follow-up empowers the hypothesis that NOAF, even if transient and related to an acute event, must be appropriately treated.

## 5. Limitations

The most important limitation of our single-center retrospective study is the absence of a control group under OAC; this feature may have influenced the reported events at the follow-up. Moreover, due to the long duration of the follow-up, we collected heterogenous treatment regimens including older therapies such as using a vitamin K antagonist. In fact, only 21.6% of patients have been prescribed OAC at hospital discharge and the long-term adverse events are limited to patients not prescribed with any of the available anticoagulants. This evidence reinforces the strength of our conclusions.

## 6. Conclusions

The STEMI patients who develop NOAF during hospitalization are at a very high-risk of death in the short- and long-term follow-up. AF should be diagnosed, treated, and converted to sinus rhythm as soon as possible, since its duration correlates with mortality. Even in the patients with a single acute episode of NOAF, our data support the use of adequate risk scores, in particular the CHA2DS2-VASc and HAS-BLED scores, to assess the indication for OAC therapy. The patients at very high risk of bleeding need a rigorous follow-up monitoring and the implantable loop recorder may be an effective tool to detect arrhythmic relapse and to identify the subjects needing a long-term anticoagulation.

## Figures and Tables

**Figure 1 diagnostics-13-00613-f001:**
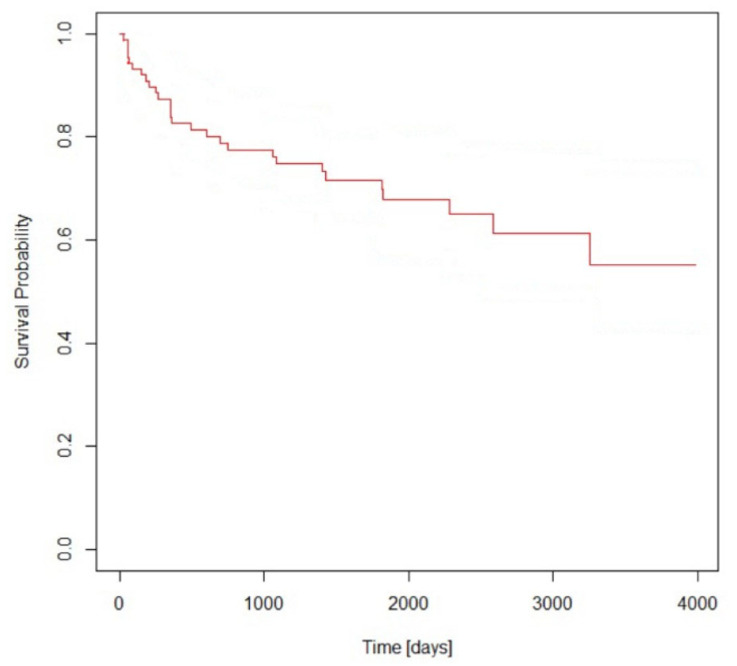
Kaplan–Meier curve for NOAF during STEMI survival.

**Table 1 diagnostics-13-00613-t001:** Clinical, procedural, arrhythmic events characteristics, and outcomes of the population.

** *Clinical Characteristics* **		
Age (M-SD)		78.8 ± 10.6
Male gender (%)		62.7
Hypertension (%)		61.8
Overweight (%)		9.8
Active Smoke (%)		28.9
Hypercolesterolemia (%)		34.3
Diabetes (%)		21.6
Killip class ≥3 (%)		17.6
Cardiogenic shock (%)		12.7
Out of hospital CA (%)		5.9
Renal failure (eGFR < 60mL/min/1.73 m^2^) (%)		40.2
eGFR (mL/min/1.73 m^2^) (M-SD)		65.3 ± 27.6
Anemia (%)		33.3
Haemoglobin (g/dL) (M-SD)		13.6 ± 2.3
Left Ventricular Ejection Fraction (%) (M-SD)		43.5 ± 12.1
Left atrial volume (mL) (M-SD)		58.0 ± 20.9
Anterior myocardial infarction (%)		46
Multivessel disease (%)		51
** *Arrhytmic Event Characteristics* **		
AF duration (hours) (M-SD)		8.1 ± 12.5
Spontaneous remission (%)		2.5
Pharmacological cardioversion (%)		69.6
Electrical cardioversion (%)		2
Persistent (%)		5.9
Anticoagulant therapy at discharge (%)		21.6
CHA_2_DS_2_-VASc Score ≥2 (%)		83.3
HAS-BLED Score ≤3 (%)		85.6
** *Outcome* **		
In-hospital mortality (%)		14.7
1-year mortality (%)		17.2
Long term mortality (%)		32.1
Stroke 1 year (%)		3.4
Bleeding 1 year (%)		-
** *Mortality Predictors* **		
** *In hospital* **	OR [95% CI]	p value
Age	1.006 [1.000–1.013]	0.05
LVEF	0.993 [0.987–0.999]	0.03
** *1-year follow-up* **		
Age	1.019 [1.011–1.026]	<0.0001
Arrhythmia duration	1.0007 [0.999–1.001]	0.04
** *Long-term follow-up* **		
Age	1.021 [1.012–1.030]	<0.0001

## Data Availability

The data presented in this study are available on request from the corresponding author.

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
