# Peer review of "New Onset Atrial Fibrillation in STEMI Patients: Main Prognostic Factors and Clinical Outcome"

_diagnostics, 2023, doi:10.3390/diagnostics13040613_

Round 1

Reviewer 1 Report

I had the opportunity to review the manuscript entitled "New onset atrial fibrillation in STEMI patients: main prognostic factors and clinical outcome".

Detailed comments:

1) The paper is not particularly well written. The text is sometimes challenging to understand. The manuscript presentation should be improved. Please, correct typos and grammatical errors. For example, in the sentence "Mortality, primarily in ST-segment elevation myocardial infarction (STEMI), remains consistent...", the term "consistent" is not appropriate.

2) Please define all abbreviations since their first mention. For example, AF and NOAF are not defined in the abstract.

3) The single-center, retrospective design of the study represents a relevant limitation that should be carefully discussed in the manuscript.

4) Methods. The authors should specify what is the primary endpoint of the study and then the additional secondary endpoints.

5) Tables 1 and 2 should include information on (i) the overall population (N=1455), (ii) patients with NOAF and (iii) patients without NOAF, separately. These tables should present the comparisons of the baseline and procedural characteristics between NOAF vs non-NOAF patients (with p-values), which are of potential interest.

6) In the Methods section, the authors report that the radial artery was the preferred approach to perform coronary angiography and PCI. This is, however, inconsistent with the 61.8% of patients treated via the femoral access reported in Table 2. Please, clarify this concept and modify the text accordingly.

7) The majority of patients discharged on long-term oral anticoagulation received warfarin. This is in contrast with the contemporary approach of preferring DOACs in these patients. This aspect may represent an additional limitations that should be discussed in the manuscript.

8) No bleeding events were collected at 1 year follow-up. This is unexpected considering the advanced age of the study cohort (mean age: 75 years) and the use of dual or triple antithrombotic therapy for STEMI/AF. The study population can be, therefore, defined at high bleeding risk and it is surprising that none of the patients experienced any minor/major bleeding. The lack of bleeding is probably due to underreporting in a retrospective analysis. The authors should double check the data and discuss this aspect in the manuscript.

9) The authors evaluate the independent predictors of mortality in NOAF patients. It would be of interest to assess whether "NOAF" is an independent predictor of fatal or nonfatal adverse events in the total population of 1455 STEMI patients. This analysis would provide additional information on the prognostic effect of NOAF in STEMI, which remains unclear.

10) The risk of bleeding is high in patients with AF undergoing PCI, particularly in the case of ACS at presentation compared with CCS (EuroIntervention. 2021;17(11):e898-e909; PMID: 34105513). I recommend briefly discussing this concept in the manuscript.

11) The graphical presentation should be improved.

Author Response

The reviewer's comments have been highly appreciated and very useful in the revision of the manuscript 

Reviewer 2 Report

I read with interest the manuscript (ID diagnostics-2091251) entitled New onset atrial fibrillation in STEMI patients: main prognostic factors and clinical outcome" by Dal Zotto et al.

They performed single-center, retrospective, cohort study on consecutive patients  undergoing coronary angiography and/or PCI for STEMI who 73 developed NOAF during the hospitalization. End-points were: in-hospital and 1-year and long-term follow-up mortality, both with cerebral ischemic and hemorrhagic events at 1 year. The results are in line with expectations.

My comments:

The sample of enrolled patients (STEMI + new onset atrial fibrillation) is small to achieve conclusive results.

Since the initial cohort of STEMI subjects is large, I suggest performing a case-control study by selecting 1 to 3 controls (STEMI without AF), matched by sex and age, for each STEMI and AF patient. This would allow for a better understanding of whether AF really represents a risk factor for worse in-hospital and long-term mortality. Obviously this makes it necessary to rewrite the study, but improving it enormously. As such the study is of much less relevant and inconclusive significance.

Too many tables/figures are shown in the manuscript. I suggest removing tables 1 and 2. Table 1 could be added as a supplementary file and the most relevant results of table 2 could be reported in the text. Figure 1 without a comparison group is not very helpful.

Author Response

(The authors gave the same response as above.)

Round 2

Reviewer 1 Report

I have no further comments.

Reviewer 2 Report

The authors responded to my comments. Too bad they couldn't change the methodology by introducing the control group. This would have given greater strength to their results, which, however, remain of moderate interest.